# Multi-Task Temporal Shift Attention Networks for On-Device Contactless Vitals Measurement

**Xin Liu[1], Josh Fromm[3], Shwetak Patel[1], Daniel McDuff[2]**
Paul G. Allen School of Computer Science & Engineering, University of Washington, Seattle, USA[1]
Microsoft Research AI, Redmond, USA[2]
OctoML, Seattle, USA[3]
{xliu0, shwetak}@cs.washington.edu, jwfromm@octoml.ai, damcduff@microsoft.com

## Abstract

Telehealth and remote health monitoring have become increasingly important during the SARS-CoV-2 pandemic and it is widely expected that this will have a lasting impact on healthcare practices. These tools can help reduce the risk of exposing patients and medical staff to infection, make healthcare services more accessible, and allow providers to see more patients. However, objective measurement of vital signs is challenging without direct contact with a patient. We present a video-based and on-device optical cardiopulmonary vital sign measurement approach. It leverages a novel multi-task temporal shift convolutional attention network (MTTS-CAN) and enables real-time cardiovascular and respiratory measurements on mobile platforms. We evaluate our system on an Advanced RISC Machine (ARM) CPU and achieve state-of-the-art accuracy while running at over 150 frames per second which enables real-time applications. Systematic experimentation on large benchmark datasets reveals that our approach leads to substantial (20%-50%) reductions in error and generalizes well across datasets.

## 1 Introduction

The SARS-CoV-2 (COVID-19) pandemic is transforming the face of healthcare around the world [1, 2]. One example of this is the sharp increase (by more than 10x) in the number of medical appointments held via telehealth platforms because of the increased pressures on healthcare systems, the desire to protect healthcare workers and restrictions on travel [2]. Telehealth includes the use of telecommunication tools, such as phone calls and messaging, and online health portals that allow patients to communicate with their providers. The Center for Disease Control and Prevention is recommending the "use of telehealth strategies when feasible to provide high-quality patient care and reduce the risk of COVID-19 transmission in healthcare settings"[1]. Performing primary care visits from a patient's home reduces the risk of exposing people to infections, increases the efficiency of visits and facilitates care for people in remote locations or who are unable to travel. These are longstanding arguments for telehealth and will still be valid after the end of the current pandemic. Healthcare systems are likely to maintain a high number of telehealth appointments in the future [3].

However, despite the longstanding promise of telehealth, it is difficult to provide a similar level of care on a video call as during an in-person visit. The physician *can* diagnose a patient based on visual observations and self-reported symptoms; however, in most cases they *cannot* objectively assess the patient's physiological state. This means that physicians have to make decisions (e.g., recommending a trip to the emergency department) without important data. In the case of COVID-19, there are severe cardiopulmonary (heart and lung related) symptoms that are difficult to evaluate remotely. The

---

Github Link: `https://github.com/xin71/MTTS-CAN`
[1]https://www.cdc.gov/coronavirus/2019-ncov/hcp/ways-operate-effectively.html

symptoms seen in patients have drawn links to acute respiratory distress syndrome [4], myocardial injury, and chronic damage to the cardiovascular system. Experts suggest that particular attention should be given to cardiovascular protection during treatment [5]. The development of more accurate and efficient non-contact cardiopulmonary measurement technology would give remote physicians access to the data to make more informed decisions. Beyond telehealth, the same technology could impact passive health monitoring, improving the standard of care for infants in neonatal intensive care units [6].

Cameras can be used to measure physiological signals, including heart and respiration rates, and blood oxygenation levels [7, 8, 9], based on facial videos [10, 11]. Non-contact cardiopulmonary measurement involves capturing subtle changes in light reflected from the body caused by physiological processes. Imaging methods can be used to measure volumetric changes of blood in the surface of the skin cause changes in light absorption ($\uparrow$ volume of hemoglobin = $\uparrow$ light absorption). This in turns affects the amount of visible light reflected from the skin, which is the source of the photo-plethysmogram (PPG). The mechanical force of blood pumping around the body also causes subtle motions and these are the source of the ballistocardiogram (BCG). These color and motion changes in the video help us extract the pulse signal and heart rate frequency. The PPG and BCG signals provide complementary information to one another and also contain information about breathing due to respiratory sinus arrhythmia [12]. Respiratory signals can also be recovered from motion-based analyses of the head and torso as the subjects breathes in and out [13].

Computer vision for remote cardiopulmonary measurement is a growing field; however, there is room for improvement in the existing methods. First, accuracy of measurements is critical to avoid false alarms or misdiagnoses. The US Federal Drug Administration (FDA) mandates that testing of a new device for cardiac monitoring should show "substantial equivalence" in accuracy with a legal predicate device (e.g., a contact sensor)[2]. This standard has not been obtained in non-contact approaches. Second, designing models that run on-device helps reduce the need for high-bandwidth Internet connections making telehealth more practical and accessible. Moreover, camera-based cardiopulmonary measurement is a highly privacy sensitive application. These data are personally identifiable, combining videos of a patient's face with sensitive physiological signals. Therefore, streaming and uploading data to the cloud to perform analysis is not ideal. Finally, the ability to run at a high frame rates enables opportunistic sensing (e.g., obtaining measurements each time you look at your phone) and helps capture waveform dynamics that could be used to detect arterial fibrillation [14], hypertension [15], and heart rate variability [16] where high-frame rates (at least 100Hz) are a requirement to yield precise measurements.

We propose a novel multi-task temporal shift convolutional attention network (MTTS-CAN) to address the challenges of privacy, portability, and precision in contactless cardiopulmonary measurement. Our end-to-end MTTS-CAN leverages temporal shift modules to perform efficient temporal modeling and remove various sources of noise without any additional computational overhead. An attention module improves signal source separation, and a multi-task mechanism shares the intermediate representations between pulse and respiration to jointly estimate both simultaneously. By combining these three techniques, our proposed network can run on an ARM CPU and achieve the state-of-the-art accuracy and inference speed.

To summarize, the contributions of this paper are to 1) present an accurate and efficient approach to perform on-device real-time spatial-temporal modeling of vitals signal, 2) evaluate our system and show **state-of-the-art performance** on two large public datasets, 3) provide an implementation of core tensor operations required for MTTS-CAN using a modern deep learning compiler and an on-device latency evaluation across different architectures showing MTTS-CAN can run at more than **150 frame** per second. Our code, models, and video figures are provided in the supplementary materials.

## 2 Related Work

**Camera-based Physiological Measurement:** Early work established that the blood volume pulse can be extracted by analysing skin pixel intensity changes over time [10, 11]. These methods are grounded by optical models (e.g., the Lambert-Beer law (LBL) and Shafer's dichromatic reflection

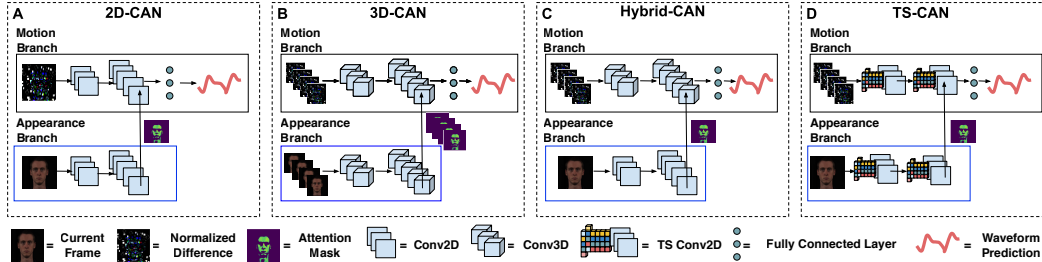

Figure 1: We perform a systematic comparison of several convolutional attention network (CAN) architectural designs. Starting from previous work that presented a 2D-CAN [31], we introduce a fully 3D-CAN, a 2D-3D Hybrid CAN in which the appearance branch takes a single frame, and our proposed temporal shift CAN. Each of these models can be applied in a single or multi-task manner.

model (DRM)) that provide a framework for modeling how light interacts with the skin. However, traditional signal processing techniques are quite sensitive to noise from other sources in video data, including head motions and illumination changes [7, 12]. To help address these issues, some approaches incorporate prior knowledge about the physical properties of the patient's skin [17, 18]. Although effective, these handcrafted signal processing pipelines make it difficult to capture the complexity of the spatial and temporal dynamics of physiological signals in video. Neural network based approaches have been successfully applied using the BVP or respiration as the target signal [9, 19, 20, 21], but these methods still struggle with effectively combining spatial and temporal information while maintaining a low computational budget. More recently, researchers have investigated on-device remote camera-based heart rate variability measurement using facial videos from smartphone cameras [22]. However, their proposed architecture requires approximately 200ms per frame inference, which is insufficient for real-time performance, and was not evaluated on public datasets.

**Efficient Temporal Models:** Yu et al. [20] have shown that applying 3D convolutional neural networks (CNNs) significantly improves performance and achieves better accuracy compared to using a combination of 2D CNNs and recurrent neural networks. The benefit of 3D CNNs implies that incorporating temporal data in all layers of the model is necessary for high accuracy. However, direct temporal modeling with 3D CNNs requires dramatically more compute and parameters than 2D based models. In addition to reducing computational cost, there are several reasons that it is highly desirable to be able to have efficient non-contact physiological measurement models that run on-device. Temporal Shift Modules [23] provide a clever mechanism that can be used to replace 3D CNNs without reducing accuracy and requiring only the computational budget of a 2D CNN. This is achieved by shifting the tensor along the temporal dimension, facilitating information exchange across multiple frames. TSM has been evaluated on the tasks of video recognition and video object detection and achieved superior performance in both latency and accuracy. Xiao et al. [24] used pretrained TSM-based residual networks as a backbone followed by two attention modules for reasoning about human-object interactions. The differences between this aforementioned work and ours is they applied attention modules as the head followed by pretrained TSM-based residual feature maps while our work applies two attention modules to the intermediate feature maps generated from regular 2D CNNs with TSM.

**Machine Learning and COVID-19:** Researchers have explored the use of machine learning from various perspectives to help combat COVID-19 [25]. Recent studies have shown that applying convolutional neural networks to CT scans can help extract meaningful radiological features for COVID-19 diagnosis and facilitate automatic pulmonary CT screening as well as cough monitoring [26, 27, 28, 29]. Researchers have also looked at the correlation between resting heart rate generated from wearable sensors and COVID-19 related symptoms and behaviors at population scale [30].

## 3 Method

### 3.1 Optical Model

For our optical basis we start with Shafer's Dichromatic Reflection Model (DRM), as in prior work [18, 9]. Specifically, we aim to capture both spatial and temporal changes and the relationship

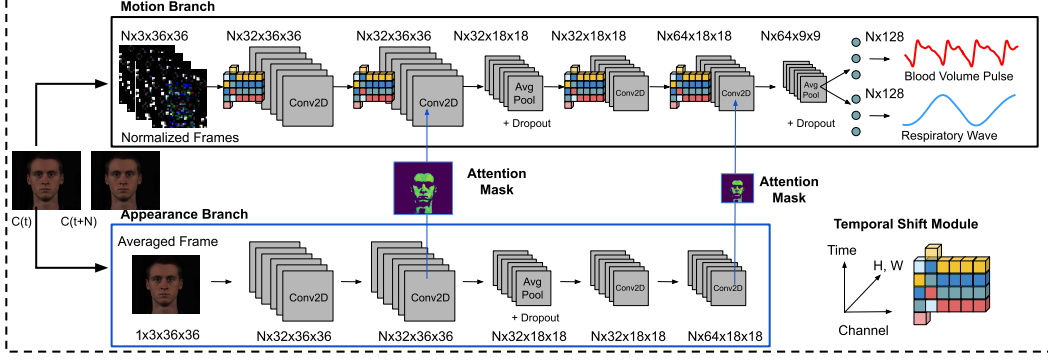

Figure 2: We present a multi-task temporal shift convolutional attention network for camera-based physiological measurement.

between multiple physiological processes. Let us start with the RGB values captured by the cameras as given by:

$$\boldsymbol{C}_k(t) = I(t) \cdot (\boldsymbol{v}_s(t) + \boldsymbol{v}_d(t)) + \boldsymbol{v}_n(t) \tag{1}$$

where $I(t)$ is the luminance intensity level, modulated by the specular reflection $\boldsymbol{v}_s(t)$ and the diffuse reflection $\boldsymbol{v}_d(t)$. The quantization noise of the camera sensor is captured by $\boldsymbol{v}_n(t)$. Following [18] we can decompose $I(t)$, $\boldsymbol{v}_s(t)$ and $\boldsymbol{v}_d(t)$ into stationary and time-dependent parts:

$$\boldsymbol{v}_d(t) = \boldsymbol{u}_d \cdot d_0 + \boldsymbol{u}_p \cdot p(t) \tag{2}$$

where $\boldsymbol{u}_d$ is the unit color vector of the skin-tissue; $d_0$ is the stationary reflection strength; $\boldsymbol{u}_p$ is the relative pulsatile strengths caused by hemoglobin and melanin absorption; $p(t)$ represents the physiological changes.

$$\boldsymbol{v}_s(t) = \boldsymbol{u}_s \cdot (s_0 + \Phi(m(t), p(t))) \tag{3}$$

where $\boldsymbol{u}_s$ denotes the unit color vector of the light source spectrum; $s_0$ and $\Phi(m(t), p(t))$ denote the stationary and varying parts of specular reflections; $m(t)$ denotes all the non-physiological variations such as flickering of the light source, head rotation, and facial expressions.

$$I(t) = I_0 \cdot (1 + \Psi(m(t), p(t))) \tag{4}$$

where $I_0$ is the stationary part of the luminance intensity, and $I_0 \cdot \Psi(m(t), p(t))$ is the intensity variation observed by the camera. As in [9] we can disregard products of time-varying components as they are relatively small:

$$\boldsymbol{C}_k(t) \approx \boldsymbol{u}_c \cdot I_0 \cdot c_0 + \boldsymbol{u}_c \cdot I_0 \cdot c_0 \cdot \Psi(m(t), p(t)) + \\ \boldsymbol{u}_s \cdot I_0 \cdot \Phi(m(t), p(t)) + \boldsymbol{u}_p \cdot I_0 \cdot p(t) + \boldsymbol{v}_n(t) \tag{5}$$

However, unlike in previous work which modeled pulse and respiration signals as independent [31], we leverage the fact that *p(t)* actually captures a complex combination of both pulse and respiration information. Specifically, both the specular and diffuse reflections are influenced by related physiological processes. Respiratory sinus arrhythmias (RSA) are rhythmical fluctuations in heart periods at the respiration frequency [32]. Furthermore, the respiration and pulse signals both cause outward motions of the body in the form of chest and head motions. We can say that the physiological process *p(t)* is a complex combination of both the blood volume pulse, *b(t)*, and the respiration wave, *r(t)*. Thus, $p(t) = \Theta(b(t), r(t))$ and the following equation gives a more accurate representation of the underlying process:

$$\boldsymbol{C}_k(t) \approx \boldsymbol{u}_c \cdot I_0 \cdot c_0 + \boldsymbol{u}_c \cdot I_0 \cdot c_0 \cdot \Psi(m(t), \Theta(b(t), r(t))) + \\ \boldsymbol{u}_s \cdot I_0 \cdot \Phi(m(t), \Theta(b(t), r(t))) + \boldsymbol{u}_p \cdot I_0 \cdot p(t) + \boldsymbol{v}_n(t) \tag{6}$$

Since *b(t)* and *r(t)* are so closely intertwined, a temporal multi-task learning approach would seem optimal for this problem and at very least could leverage redundancies between the two signals.

## 3.2 Architecture

**Efficient Spatial-Temporal Modeling:** To achieve state-of-the-art performance in on-device optical cardiopulmonary measurement, an architecture should have the ability to: 1) efficiently learn spatial features that map raw RGB values to latent representations corresponding to the pulse and respiratory signals as well as temporal features that offset various sources of noise (e.g., head motion, ambient illumination changes, etc.), 2) learn the relationships between associated physiological processes, 3) work in real-time to support various telehealth deployments. Our solution is a novel temporal shift convolutional attention architecture (Fig. 1D) which we systematically compare to its variants (Fig. 1A-C) to illustrate its benefits.

Because of the strong performance shown in prior work [9], our architecture leverages a two-branch structure with a spatial attention module (Fig. 1A). One branch is used for motion modeling, and the other branch for extracting meaningful spatial (i.e., facial) features. However, it fails to capture temporal dependencies beyond consecutive frames and thus is still vulnerable to many sources of noise. Perhaps the simplest way to introduce a strong temporal dependency is a *3D-CAN* that leverages 3D convolutions to model temporal relationships (Fig. 1B) which is similar to the model used in [20] but adds an attention module. However, since 3D convolutions incur quadratic computational cost compared to 2D convolutions, it is not feasible to achieve real-time on-device performance using a primarily 3D architecture. Therefore, we present a *Hybrid-CAN* architecture that is more computationally efficient than a purely 3D model. *Hybrid-CAN* combines a 2D-CAN and a 3D-CAN to maintain temporal modeling while leveraging more efficient 2D convolutions where possible. Since spatial position changes between adjacent frames are subtle, using 3D convolutions in the appearance branch is unnecessary. As Fig. 1C illustrates, the input of the appearance branch is a single frame generated by averaging N (window size) adjacent frames. Although Hybrid-CAN reduces computational cost significantly, the computational overhead from 3D convolutions in the motion branch is still not tolerable if we want to achieve real-time inference on low-end mobile platforms (i.e., ideally at least 60 FPS).

Therefore, we introduce *TS-CAN* to remove the 3D convolution operations from the architecture entirely while preserving spatial-temporal modeling. TS-CAN has two major additional components: the temporal shift module (TSM) [23] and the attention module. TSM performs tensor shifting before the tensor is fed into the convolutional layers as visualized in Fig.2. More specifically, TSM splits the input tensor into three chunks across the channel dimension. Then, it shifts the first chunk to the left by one place (advancing time by one frame) and shifts the second chunk to the right by one place (delaying time by one frame). Both shifting operations are along the temporal axis, and the third chunk remains unchanged. It is worth noting that tensor shifting does not add any additional parameters to the network, but does enable information exchange among neighbouring frames. We used TSM in the motion branch to mimic the effects of 3D convolution, while the appearance branch in the TS-CAN is the same as Hybrid-CAN and only takes a single averaged frame. By doing so, the model not only significantly reduces computational time by only calculating the attention mask once, but also captures most of the pixels that contain skin and reduces camera quantization error.

**Attention on Temporal Shift:** Given there are already many different sources of noise described in the previous section, naively shifting an input tensor in time will introduce extra temporal information to our representation. It is then important that we pay attention to the pixels with physiological signals or risk amplifying noise. Therefore, we propose inserting an attention module in TSM to minimize the negative effects introduced by tensor shifting as well as to enable the network to focus on the target signals. The spatial and temporal distribution of physiological signals are not uniform on human skin. Soft-attention masks can assign higher weights to certain shifted pixels with stronger signals in intermediate representations from the convolutional operations. More concretely, our attention modules are the bridges between the appearance branch and the motion branch (See Fig. 2). Softmax attention masks are generated via $1 \times 1$ convolutions before pooling layers. The attention mask is calculated as in Equation 7 where $k$ is the index of a layer, $\omega^k$ is the $1 \times 1$ convolution and followed by a sigmoid activate function $\sigma(\cdot)$. $l_1$ normalization was applied to soften the extreme values in the mask to make sure the network avoided pixel anomalies. Finally, we perform an element-wise product to the corresponding representation $\mathbb{X}^k$ from the motion branch.

$$\mathbb{X}^k \odot \frac{H_k W_k \cdot \sigma(\omega^k \mathbb{X}_\alpha^k + b^k)}{2 \parallel \sigma(\omega^k \mathbb{X}_\alpha^k + b^k) \parallel_1} \tag{7}$$

**Multi-Task TS-CAN:** We now have an efficient on-device architecture to predict physiological signals in real-time. However, we still have two independent networks, one for estimating the blood volume pulse and another for the respiration signals. Thus, the computational cost is doubled while preventing the possibility for information sharing across these related physiological processes. As we know that pulse and respiration are linked, we propose a multi-task variant of our network (see Fig. 2). This shrinks the computational budget by approximately 50% and the tasks of estimating BVP and respiration can share an intermediate representation. The loss function of this multi-task TS-CAN (MTTS-CAN) is described in Eqn. 8 where b(t) is the gold-standard BVP waveform and r(t) is gold-standard respiration waveform. b(t)' and r(t)' are the respective predictions from the model.

$$L = \frac{1}{T} \sum_{t=1}^{T} |b(t) - b(t)'| + \alpha \frac{1}{T} \sum_{t=1}^{T} |r(t) - r(t)'| \tag{8}$$

## 4  Experiments

We compare our methods to four approaches for pulse measurement: POS[18], CHROM[17], ICA[12], 2D-CAN[9] and two for respiration measurement: 2D-CAN and ARM [13]. Other than DeepPhys [9], we are not aware of other methods that work for both pulse and respiration measurement. We run our experiments using the following datasets:

**AFRL** [33]: 300 videos of 25 participants (17 males) recorded at 658x492 resolution and 120 fps (down-sampled to 30 fps for our experiments). Fingertip reflectance photoplethysmograms(PPG) were used to record ground-truth signals for training our network and electrocardiograms(ECG) were recorded for evaluating performance (this is the medical gold-standard). Each participant was recorded six times with increasing head motion in each task. The participants were asked to sit still for the first two tasks and perform three motion tasks rotating their head about the vertical axis with an angular velocity of 10 degrees/second, 20 degrees/second, 30 degrees/second, respectively. In the last task, participants were asked to orient their head randomly once every second to one of nine predefined locations. The six recording were repeated twice in front of two backgrounds.

**MMSE-HR** [34]: 102 videos of 40 participants were recorded at 1040x1392 resolution and 25 fps during spontaneous emotion elicitation experiments. The gold standard contact signal was measured via a Biopac2 MP150 system[3] which provided pulse rate at 1000 fps and was updated after each heart beat. These videos feature smaller but more spontaneous motions than those in the AFRL dataset including facial expressions. Respiration measurements were not provided.

**Experiment Details:** At a high-level all our proposed networks share a similar two-branch architecture. Each branch has four convolutional layers. There is an averaging pooling layer and dropout layer placed after the second and fourth convolutional layers as shown in Fig. 2. Different architectures in Fig. 1 require different convolutional operations (e.g., 3D-CAN requires 3D CNNs). To preprocess the input of the appearance branch, we downsample each frame $c(t)$ to 36×36, which balances maintaining spatial resolution quality and suppressing camera noise [35]. For the motion branch, we calculate normalized frames using every two adjacent frames as $(c(t+1) - c(t))/(c(t) + c(t+1))$. The normalized frames are less vulnerable to changes in brightness and skin appearance compared to the raw frames $c(t)$ and reduce the chance of over-fitting to certain datasets.

Our system is implemented in TensorFlow [36]. We trained our proposed MTTS-CAN architectures using the Adadelta optimizer [37] with a learning rate of 1.0, batch size of 32, kernel size of 3×3, pooling size of 2×2, and dropout rates of 0.25 and 0.5. The final model was chosen after the training converged (12 epochs on the respiration task and 24 epochs on the pulse task). We implemented 2D-CAN, 3D-CAN and Hybrid-CAN as baselines to compare against our proposed architectures. For the 3D and Hybrid models the training schema is similar to TS-CAN, but we use a kernel size of 3×3×3 and a pooling size of 2×2×2. We used a window size of 10 frames in all temporal models to provide a fair comparison for our proposed architectures. We picked $\alpha = 0.5$ for the multi-tasking loss function in the MTTS-CAN to force estimations of pulse and respiration treated equally (pulse and respiration signals were both normalized in amplitude). To calculate the performance metrics, we post-processed the outputs of all methods in the same way using a 2nd-order Butterworth filter (cut-off frequencies of 0.75 and 2.5 Hz for HR and 0.08 and 0.5 Hz for BR). For the AFRL data, we divided the dataset into 30-second windows with no overlap. For the MMSE-HR dataset we used a

Table 1: Pulse and respiration measurement on the AFRL and MMSE-HR datasets.

| Method | Heart Rate | | | | | | | | Respiration Rate | | | | Time |
| | AFRL (All Tasks) | | | | MMSE-HR | | | | AFRL (All Tasks) | | | | (ms) |
| | MAE | RMSE | $\rho$ | SNR | MAE | RMSE | $\rho$ | SNR | MAE | RMSE | $\rho$ | SNR | |
|---|---|---|---|---|---|---|---|---|---|---|---|---|---|
| MTTS-CAN | 1.45 | 3.72 | 0.94 | 8.64 | 3.00 | 5.66 | 0.92 | 2.37 | 2.30 | 4.52 | 0.40 | 18.7 | 6 |
| MT-Hyb.-CAN | 1.15 | 2.69 | 0.97 | 10.2 | 3.43 | 6.98 | 0.88 | 4.70 | 2.17 | 4.24 | 0.45 | 19.1 | 13 |
| TS-CAN | 1.32 | 3.25 | 0.95 | 8.86 | 3.41 | 7.82 | 0.84 | 2.92 | 2.25 | 4.47 | 0.41 | 18.9 | 12 |
| Hyb.-CAN | 1.12 | 2.60 | 0.97 | 10.6 | 2.55 | 4.16 | 0.96 | 5.47 | 2.06 | 4.17 | 0.46 | 19.8 | 26 |
| 3D-CAN | 1.18 | 2.83 | 0.97 | 10.5 | 2.78 | 5.08 | 0.94 | 4.73 | 2.31 | 4.42 | 0.44 | 19.3 | 48 |
| 2D-CAN [9] | 2.32 | 5.82 | 0.85 | 6.23 | 4.72 | 8.68 | 0.82 | 2.06 | 2.86 | 5.16 | 0.34 | 16.3 | 20 |
| POS [18] | 2.48 | 5.07 | 0.89 | 2.32 | 3.90 | 9.61 | 0.78 | 2.33 | | | | | - |
| CHROM [17] | 6.42 | 12.4 | 0.60 | -4.83 | 3.74 | 8.11 | 0.82 | 1.90 | Not Applicable | | | | - |
| ICA [12] | 4.36 | 7.84 | 0.77 | 3.64 | 5.44 | 12.00 | 0.66 | 3.03 | | | | | - |
| ARM [13] | Not Applicable | | | | | | | | 3.68 | 5.52 | 0.29 | -6.22 | - |

MAE = Mean Absolute Error, RMSE = Root Mean Squared Error, $\rho$ = Pearson Correlation, SNR = BVP Signal-to-Noise Ratio.

Table 2: Pulse and respiration measurement MAE on the AFRL by motion task.

| Method | Heart Rate | | | | | | Respiration Rate | | | | | |
| | T1 | T2 | T3 | T4 | T5 | T6 | T1 | T2 | T3 | T4 | T5 | T6 |
|---|---|---|---|---|---|---|---|---|---|---|---|---|
| MTTS-CAN | 1.08 | 1.23 | 0.94 | 1.27 | 1.07 | 3.12 | 0.68 | 0.98 | 2.12 | 3.81 | 3.31 | 2.89 |
| MT-Hybrid-CAN | 1.04 | 1.24 | 0.95 | 1.23 | 0.88 | 1.53 | 0.77 | 0.89 | 2.23 | 3.28 | 3.03 | 2.80 |
| TS-CAN | 1.07 | 1.25 | 0.96 | 1.24 | 1.01 | 2.36 | 0.69 | 1.14 | 2.27 | 3.70 | 3.18 | 2.53 |
| Hybrid-CAN | 1.04 | 1.21 | 0.94 | 1.22 | 0.89 | 1.39 | 0.77 | 1.03 | 1.83 | 3.19 | 2.96 | 2.60 |
| 3D-CAN | 1.06 | 1.19 | 0.92 | 1.23 | 0.89 | 1.77 | 0.96 | 0.98 | 2.58 | 3.80 | 2.87 | 2.65 |
| 2D-CAN [9] | 1.08 | 1.21 | 1.02 | 1.43 | 2.15 | 7.05 | 1.25 | 1.11 | 3.35 | 4.63 | 3.77 | 3.08 |
| POS [18] | 1.50 | 1.53 | 1.50 | 1.84 | 2.05 | 6.11 | | | | | | |
| CHROM [17] | 4.53 | 4.59 | 4.35 | 4.84 | 6.89 | 10.3 | Not Applicable | | | | | |
| ICA [12] | 1.17 | 1.70 | 1.70 | 4.00 | 5.22 | 11.8 | | | | | | |
| ARM [13] | Not Applicable | | | | | | 2.51 | 2.53 | 3.19 | 4.85 | 4.22 | 4.78 |

window size equal to the number of frames in each video. We then computed four standard metrics for each window: mean absolute error (MAE), root mean squared error (RMSE) and correlation ($\rho$) in heart/breathing rate estimations and the corresponding BVP/respiration signal-to-noise ratio (SNR) [17]. Details of the calculation for these metrics, training code, architecture and the trained models are available in the supplementary material.

**On-Device Evaluation:** Our proposed architectures were deployed on an open-source embedded system called Firefly-RK3399[4] for latency evaluation. This embedded system has two large Cortex-A72 cores and four small Cortex-A53 cores. Although RK3399 also has a mobile Mali GPU, we focus our evaluation on CPU such that our proposed end-to-end architecture can be generalized to any ARM based mobile platform and IoT device. In this work, we extend a deep learning compiler stack - TVM [38] to support the core temporal shift operation required for TS-CAN. TVM takes a high-level description of a function and generates highly optimized low-level code for a targeted device. More specifically, our TVM-based on-device system first converts a TensorFlow graph to a Relay graph [39] and complies the code to Firefly-RK3399 using LLVM. We take advantage of TVM's scheduling primitives to generate efficient low-level LLVM code that accelerates expensive operations such as 2D and 3D convolutions.

## 5 Results and Discussion

**Comparison with the State-of-the-Art:** For the AFRL dataset all 25 participants were randomly divided into five folds of five participants each (same folds as in [9]). The learning models were trained and tested via five-fold cross-validation using data from all tasks. The evaluation metrics are averaged over five folds and shown in Table 1. All of our proposed models outperform the 2D-CAN and other baselines. Hybrid-CAN and 3D-CAN achieve similar accuracy, reducing MAE by 50% on pulse and 20% on respiration measurement. The hybrid model has lower computational cost and is therefore preferable. TS-CAN also surpasses the 2D-CAN by more than 43% on pulse and 20% on respiration measurement. We also evaluated a multi-tasking version of TS-CAN and

Hybrid-CAN, and call them MTTS-CAN and MT-Hybrid-CAN respectively. We observe that there is no accuracy benefit from the multi-tasking model variants relative to the single task versions because the network must use almost all the same parameters for both tasks. However, the MT models require half the computation and half as many parameters compared to running pulse and respiration models separately which is a considerable benefit.

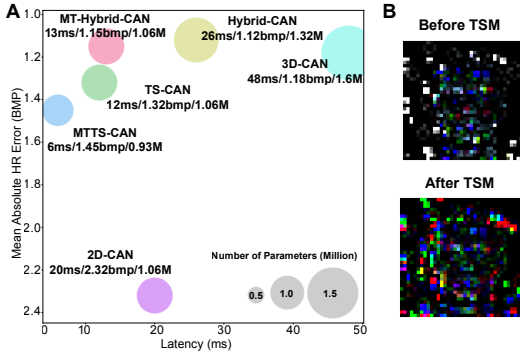

Figure 3: (A) On-Device latency evaluation across six models; (B) An visualization of TSM on a normalized frame from motion branch.

**Cross-Dataset Generalization:** To test whether our model can generalize to videos with a different resolution, background, and lighting, we trained our proposed models on the AFRL dataset and tested on the MMSE-HR dataset. Our proposed TS-CAN, Hybrid-CAN and 3D-CAN reduce errors by 25-50% compared to 2D-CAN (see Table 1). Furthermore, MTTS-CAN and MT-Hybrid-CAN both perform strongly, showing that it is possible to share the representations between pulse and respiration.

**Computation Cost and Latency:** Fig. 3A and the last column of Table 1 show that MTTS-CAN and TS-CAN are the fastest architectures of those evaluated, taking 6 ms and 12 ms per frame for inference respectively. It is worth noting that TS-CAN is 40% faster than the 2D-CAN because the unique design of the appearance branch that only executes once and provides the generated attention mask to all the frames in the motion branch. MT-Hybrid-CAN and Hybrid-CAN also achieve 13ms and 26ms inference times respectively, this is approximately double that of our TS-based methods due to the cost of 3D convolutions relative to 2D convolutions. The 2D-CAN not only has a higher latency compared to TS-CAN, but the accuracy is significantly lower. It is not surprising that the 3D-CAN achieved the worst inference speed because it has costly 3D convolutions in both branches. Latency is important because we want our models to run at as high a frame rate as possible, 30 fps is the bare minimum required to accurately measure heart rate variability and subtle waveform dynamics and 100 fps would be preferable. Therefore, faster inference increases the precision at which we can measure inter-beat and systolic-diastolic intervals [16] and could help with non-invasive blood pressure measurement [15] and detecting arterial fibrillation (AFib) [14].

**Temporal Modeling:** Capturing such waveform dynamics requires good temporal modeling, therefore we compared several designs to help improve this. Our proposed MTTS-CAN, TS-CAN, MT-Hybrid-CAN, Hybrid-CAN and 3D-CAN all outperform the 2D-CAN and other baseline methods. This is consistent with prior work that found a 3D-CNN without attention outperformed a 2D-CNN (without attention) [20]. We would anticipate that the focus on modeling the temporal aspects of the physiological waveform would lead to greater resilience to noise. We perform a systematic evaluation on videos with varying velocities of angular (rotational) head motion. The results are shown in Table 2. As expected, all the proposed temporal models perform particularly strongly on tasks with greater velocity head motion; reducing the error on the most challenging task (6) by over 75%. Moreover, as Fig. 3B illustrates, although tensor shifting provides important temporal information, it also introduces extra noise. The results in Table 1 indicate that our attention module is effective at separating the signal from the added noise.

**Multi-task Learning:** Comparing our MT models with the non-MT models, we observe that the MT models do not reduce the error in pulse and respiration rate estimates. But they do significantly improve the efficiency of inference as shown in Fig. 3A which is critical in resource constrained mobile platforms. Moreover, in order to estimate heart beat and respiration rate from a video, there is a number of mandatory pre-processing and post-processing steps to be included in the pipeline such as down-sampling images, computing averaged frames, calculating the number of peaks etc. Since MTTS-CAN only takes 6ms for inference on each fraem, even with the pre-processing overhead real-time inference is still eminently feasible. Also, memory is a valuable resource on edge devices, and MTTS-CAN only requires half of the memory to store the parameters compared to TS-CAN. We believe MTTS-CAN can be deployed and especially useful in resource constrained settings.

**Applications of MTTS-CAN:** The low latency and high accuracy of our system opens the door for many other applications. For example, it could be used to improve the measurement of heart rate variability which is a measure of the variation in the time between each heartbeat. Tracking the subtle changes between consecutive heart beats requires low latency like that provided by MTTS-CAN. Contactless and on-device HRV tracking could enable numerous novel applications in mental health and personalized health. Besides health applications, MTTS-CAN is also potentially be applied to various computer vision tasks that require on-device computation such as activity recognition and video understanding.

# 6   Broader Impact

Non-contact camera-based vital sign monitoring has great potential as a tool for telehealth. Our proposed system can promote global health equity and make healthcare more accessible for those in rural areas or those who find it difficult to travel to clinics and hospitals in-person (perhaps because of age, mobility issues or care responsibilities). These needs are likely to be particularly acute in low-resource settings. Non-contact sensing has other potential benefits for measuring the vitals of infants who ideally would not have contact sensors attached to their delicate skin. Furthermore, due to the exceptionally fast inference speed, the computational budget required for our proposed system is minimal. Therefore, people who cannot afford high-end computing devices still will be able to access the technology. While low-cost, ubiquitous sensing democratizes physiological measurement, it presents other challenges. If measurement can be performed from only a video, what happens if we detect a health condition in an individual when analyzing a video for other purposes. When and how should that information be disclosed? If the system fails in a context where a person is in a remote location, it may lead them to panic.

It is also important to consider how such technology could be used by "bad actors" or applied with negligence and without sufficient forethought for the implications. Non-contact sensing could be used to measure personal physiological information without the knowledge of the subject. Law enforcement might be tempted to apply this in an attempt to detect individuals who appear "nervous" via signals such as an elevated heart rate or irregular breathing, or an employer may surreptitiously screen prospective employees for health conditions without their knowledge during an interview. These applications would set a very dangerous precedent and would be illegal in many cases. Just as is the case with traditional contact sensors, it must be made transparent when these methods are being used and subjects should be required to consent before physiological data is measured or recorded. There should be no penalty for individuals who decline to be measured. Ubiquitous sensing offers the ability to measure signals in more contexts, but that does not mean that this should necessarily be acceptable. Just because cameras may be able to measure these signals in a new context, or with less effort, it does not mean they should be subject to any less regulation than existing sensors, in fact quite the contrary.

In the United States, the Health Insurance Portability and Accountability Act (HIPAA) and the HIPAA Privacy Rule sets a standard for protecting sensitive patient data and there should be no exception with regard to camera-based sensing. In the case of videos there should be particular care in how videos are transferred, given that significant health data can be contained with the channel. That was one of the motivations for designing our methods to run on-device, as it can minimize the risks involved in data transfer.

# 7   Conclusions

Telehealth and the SARS-CoV-2 pandemic have acutely highlighted the specific need for accurate and computationally efficient cardiovascular and pulmonary sensing. We have presented a novel multi-task temporal shift convolutional attention network (MTTS-CAN) that improves on the state-of-the-art in both of these dimensions.

# 8   Acknowledgements

We would like to thank the financial support from Bill & Melinda Gates Foundation and University of Washington Endowment Funds.

## Footnotes

[2]https://www.fda.gov/regulatory-information/search-fda-guidance-documents/cardiac-monitor-guidance-including-cardiotachometer-and-rate-alarm-guidance-industry#6_1

[3] https://www.biopac.com/

[4]`http://en.t-firefly.com/product/rk3399.html`

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
