[Supplementary Material]

# SUPPLEMENTARY MATERIAL FOR:
# Multi-Task Temporal Shift Attention Networks for On-Device Contactless Vital Measurement

**Xin Liu[1], Josh Fromm[3], Shwetak Patel[1], Daniel McDuff[2]**

Paul G. Allen School of Computer Science & Engineering, University of Washington, Seattle, USA[1]
Microsoft Research AI, Redmond, USA[2]
OctoML, Seattle, USA[3]
{xliu0, shwetak}@cs.washington.edu, jwfromm@octoml.ai, damcduff@microsoft.com

**We have an associated video attached on the supplementary material.**

## 1 Example Waveforms

Please see our video figure in the supplementary material for video examples of results for pulse and respiration results. Below we should examples of pulse (Fig. 1) and respiration (Fig. 2) waveforms for the MT-2DCAN, MT-Hybrid-CAN and MTTS-CAN models compared to the gold-standard contact sensor measurements. Videos are shown in the video figure included with the supplementary material.

Figure 1: Examples of pulse wave predictions from MT-2DCAN, MT-Hybrid-CAN and MTTS-CAN. Notice how the agreement with the gold-standard measurements (blue) is highest for MT-Hybrid-CAN and MTTS-CAN.

Figure 2: Examples of respiration wave predictions from MT-2DCAN, MT-Hybrid-CAN and MTTS-CAN. Notice how the agreement with the gold-standard measurements (blue) is highest for MTTS-CAN.

## 2 Architecture Details

- Please see Fig. 3 for details of TS-CAN and MTTS-CAN.
- Please see Fig. 4 for details of Hybrid-CAN and MT-Hybrid-CAN.
- Please see Fig. 5 for details of 3D-CAN and MT-3D-CAN.

## 3 Pre-processing

All the input images are first downsampled to the resolution of $36 \times 36$. We then calcucated the normalized frames between two consecutive frames using $(c(t+1) - c(t))/(c(t+1) + c(t))$ to reduce the dependency on the absolute frame brightness and skin appearance. By doing so, we have a normalized image and a raw image for every single frame in the video. Both of raw and normalized frame were mean subtracted and then scaled to unit standard deviation over each video.

## 4 Post-processing

To calculate the performance metrics, we post-processed the outputs of all methods in the same way using a 2nd-order Butterworth filter (cut-off frequencies of 0.75 and 2.5 Hz for HR and 0.08 and 0.5 Hz for BR). For the AFRL data, we divided the dataset into 30-second windows with no overlap. For the MMSE-HR dataset we used a window size equal to the number of frames in each video. We then computed four standard metrics for each window: mean absolute error (MAE), root mean squared error (RMSE) and correlation ($\rho$) in heart/breathing rate estimations and the corresponding BVP/respiration signal-to-noise ratio (SNR).

The performance metrics are calculated as follows:

**Mean Absolute Error (MAE)** For calculate the MAE between our model estimates and the gold-standard heart rate calculated from the contact sensor measurements in window within each dataset. As follows:

**Heart Rate:**

$$MAE = \frac{1}{T} \sum_{i=1}^{T} |HR_i - HR'_i| \tag{1}$$

**Respiration Rate:**

$$MAE = \frac{1}{T} \sum_{i=1}^{T} |RR_i - RR'_i| \tag{2}$$

**Root Mean Squared Error (RMSE)** For calculate the RMSE between our model estimates and the gold-standard heart rate calculated from the contact sensor measurements in window within each dataset. As follows:

**Heart Rate:**

$$RMSE = \sqrt{\frac{i=1}{T} \sum_{1}^{T} (HR_i - HR'_i)^2} \tag{3}$$

**Respiration Rate:**

$$RMSE = \sqrt{\frac{i=1}{T} \sum_{1}^{T} (RR_i - RR'_i)^2} \tag{4}$$

Where HR and RR are the gold-standard heart rate and respiration rates and HR' RR' are the estimated heart rate and respiration rates, respectively, from the video. The gold-standard HR frequency was determined from the manually corrected ECG peaks in the AFRL dataset and the HR estimates provided with the dataset for the MMSE-HR dataset.

We also compute the Pearson correlation between the estimated heart rates and respiration rates and the gold-standard heart rates from the contact sensor measurements.

**Pulse and Respiration Signal-to-Noise Ratios (SNR):**

We calculate blood volume pulse and respiration signal-to-noise ratios (SNR) according to the method proposed by De Haan et al. (De Haan & Jeanne, 2013). This captures the signal quality of the recovered pulse and respiration estimates. Again, the gold-standard HR/RR frequency was determined from the manually corrected gold-standard measurements in the AFRL and MMSE-HR datasets, respectively.

**BVP SNR:**

$$SNR = 10\log_{10}\left(\frac{\sum_{f=30}^{240}((U_t(f)\hat{S}(f))^2}{\sum_{f=30}^{240}(1-U_t(f))\hat{S}(f))^2)}\right) \qquad (5)$$

where $\hat{S}$ is the power spectrum of the BVP signal (S), $f$ is the frequency (in BPM), HR is the heart rate computed from the gold-standard device and $U_t(f)$ is a binary template that is one for the heart rate region from HR-6 BPM to HR+6BPM and its first harmonic region from 2*HR-12BPM to 2*HR+12BPM, and 0 elsewhere.

**Respiration SNR:**

$$SNR = 10\log_{10}\left(\frac{\sum_{f=5}^{30}((U_t(f)\hat{S}(f))^2}{\sum_{f=5}^{30}(1-U_t(f))\hat{S}(f))^2)}\right) \qquad (6)$$

where $\hat{S}$ is the power spectrum of the respiration signal (S), $f$ is the frequency (in breaths/min), RR is the respiration rate computed from the gold-standard device and $U_t(f)$ is a binary template that is one for the heart rate region from RR-6 breaths/min to RR+6breaths/min and its first harmonic region from 2*RR-12breaths/min to 2*RR+12breaths/min, and 0 elsewhere.

## 5  Baseline Methods

We compared the performance of our proposed approach to state-of-the-art supervised method using a convolutional attention network (CAN) and three unsupervised methods described below.

For the CHROM, ICA and POS methods face detection was first performed using MATLAB's face detection (`vision.CascadeObjectDetector()`). This was fixed for all methods, to avoid the influence of the face detector on performance. For the CAN methods following the implementation in Chen & McDuff (2018) we did not use face detection but rather we passed the full frame to the network after cropping the center portion to make the frame a square with W=H.

**CHROM** (De Haan & Jeanne, 2013). This method uses a linear combination of the chrominance signals obtained from the RGB video. The $[x_R, x_G, x_B]$ signals are filtered using a zero-phase, 3rd-order Butterworth bandpass filter with pass-band frequencies of [0.7 2.5] Hz. Following this, a moving window method of length 1.6 seconds (with overlapping windows and a step size of 0.8 seconds) is applied. Within each window the color signals are normalized by dividing by their mean value to give $[\bar{x}_r, \bar{x}_g, \bar{x}_b]$. These signals are bandpass filtered using zero-phase forward and reverse 3rd-order Butterworth filters with pass-band frequencies of [0.7 2.5] Hz. The filtered signals $[y_r, y_g, y_b]$ are then used to calculate $S_{win}$:

$$S_{win} = 3(1 - \frac{\alpha}{2})y_r - 2(1 + \frac{\alpha}{2})y_g + \frac{3\alpha}{2}y_b \qquad (7)$$

Where $\alpha$ is the ratio of the standard deviations of the filtered versions of A and B:

$$A = 3y_r - 2y_g \qquad (8)$$

$$B = 1.5y_r + y_g - 1.5y_b \qquad (9)$$

The resulting outputs are scaled using a Hanning Window and summed with the subsequent window (with 50% overlap) to construct the final blood volume pulse (BVP) signal.

**ICA** (Poh et al., 2010). This approach involves spatial averaging the pixels by color channel in the region of interest (ROI) for each frame to form time varying signals $[x_R, x_G, x_B]$. Following this, the observation signals are detrended. A Z-transform is applied to each of the detrended signals. The

Table 1: Pulse and respiration measurement on the AFRL and MMSE-HR datasets.

| | Heart Rate | | | | | | | | Respiration Rate | | | | Time |
| | AFRL (All Tasks) | | | | MMSE-HR | | | | AFRL (All Tasks) | | | | |
| Method | MAE | RMSE | $\rho$ | SNR | MAE | RMSE | $\rho$ | SNR | MAE | RMSE | $\rho$ | SNR | (ms) |
|---|---|---|---|---|---|---|---|---|---|---|---|---|---|
| MTTS-CAN | 1.45 | 3.72 | 0.94 | 8.64 | 3.00 | 5.66 | 0.92 | 2.37 | 2.30 | 4.52 | 0.40 | 18.7 | 6 |
| MT-Hyb.-CAN | 1.15 | 2.69 | 0.97 | 10.2 | 3.43 | 6.98 | 0.88 | 4.70 | 2.17 | 4.24 | 0.45 | 19.1 | 13 |
| MT-3D-CAN | 1.20 | 2.82 | 0.97 | 10.4 | 3.70 | 7.85 | 5.35 | 0.84 | 2.21 | 4.37 | 0.43 | 18.8 | 24 |
| MT-2D-CAN | 2.51 | 6.20 | 0.83 | 6.03 | 5.14 | 10.3 | 0.73 | 1.83 | 2.98 | 5.23 | 0.33 | 16.2 | 10 |
| TS-CAN | 1.32 | 3.25 | 0.95 | 8.86 | 3.41 | 7.82 | 0.84 | 2.92 | 2.25 | 4.47 | 0.41 | 18.9 | 12 |
| Hyb.-CAN | 1.12 | 2.60 | 0.97 | 10.6 | 2.55 | 4.16 | 0.96 | 5.47 | 2.06 | 4.17 | 0.46 | 19.8 | 26 |
| 3D-CAN | 1.18 | 2.83 | 0.97 | 10.5 | 2.78 | 5.08 | 0.94 | 4.73 | 2.31 | 4.42 | 0.44 | 19.3 | 48 |
| 2D-CAN | 2.32 | 5.82 | 0.85 | 6.23 | 4.72 | 8.68 | 0.82 | 2.06 | 2.86 | 5.16 | 0.34 | 16.3 | 20 |

MAE = Mean Absolute Error, RMSE = Root Mean Squared Error, $\rho$ = Pearson Correlation, SNR = BVP Signal-to-Noise Ratio.

Table 2: Pulse and respiration measurement MAE on the AFRL by motion task.

| | Heart Rate | | | | | | Respiration Rate | | | | | |
| Method | T1 | T2 | T3 | T4 | T5 | T6 | T1 | T2 | T3 | T4 | T5 | T6 |
|---|---|---|---|---|---|---|---|---|---|---|---|---|
| MTTS-CAN | 1.08 | 1.23 | 0.94 | 1.27 | 1.07 | 3.12 | 0.68 | 0.98 | 2.12 | 3.81 | 3.31 | 2.89 |
| MT-Hybrid-CAN | 1.04 | 1.24 | 0.95 | 1.23 | 0.88 | 1.53 | 0.77 | 0.89 | 2.23 | 3.28 | 3.03 | 2.80 |
| MT-3D-CAN | 1.07 | 1.21 | 0.92 | 1.23 | 0.86 | 1.91 | 0.81 | 1.00 | 2.10 | 3.33 | 3.05 | 3.00 |
| MT-2D-CAN | 1.08 | 1.28 | 1.13 | 1.52 | 2.60 | 7.46 | 1.47 | 1.24 | 3.77 | 4.53 | 3.58 | 3.32 |
| TS-CAN | 1.07 | 1.25 | 0.96 | 1.24 | 1.01 | 2.36 | 0.69 | 1.14 | 2.27 | 3.70 | 3.18 | 2.53 |
| Hybrid-CAN | 1.04 | 1.21 | 0.94 | 1.22 | 0.89 | 1.39 | 0.77 | 1.03 | 1.83 | 3.19 | 2.96 | 2.60 |
| 3D-CAN | 1.06 | 1.19 | 0.92 | 1.23 | 0.89 | 1.77 | 0.96 | 0.98 | 2.58 | 3.80 | 2.87 | 2.65 |
| 2D-CAN | 1.08 | 1.21 | 1.02 | 1.43 | 2.15 | 7.05 | 1.25 | 1.11 | 3.35 | 4.63 | 3.77 | 3.08 |

Independent Component Analysis (ICA) (JADE implementation) is applied to the normalized color signals.

**POS** (Wang et al., 2016). The intensity signals $[x_R, x_G, x_B]$ are computed. A moving window of length 1.6 seconds (with overlapping windows and with a step size of one frame), is applied. For each time window, the signal is divided by its mean to give $[\bar{x_r}, \bar{x_g}, \bar{x_b}]$. Following this, $X_s$ and $Y_s$ are calculated where:

$$X_s = \bar{x}_g - \bar{x}_b \tag{10}$$

$$Y_s = -2\bar{x}_r + \bar{x}_g + \bar{x}_b \tag{11}$$

$X_s$ and $Y_s$ are then used to calculate $S_{win}$, where:

$$S_{win} = X_s + \frac{\sigma(X_s)}{\sigma(Y_s)}Y_s \tag{12}$$

The resulting outputs of the window-based analysis are used to construct the final BVP signal in an overlap add fashion.

# 6   Additional Experimental Results

Table 1 and Table 2 shows the results of all the multi-task and non-multi-task architectures. Including the MT-3DCAN and MT-2DCAN for which we did not have space in the main paper.

# 7   Additional Figures of Results

Figure 6 and Figure 7 demonstrate the relationship and distribution between estimated HR and ground-truth HR across all the subjects in the AFRL dataset. Figure 8 and Figure 9 demonstrate the relationship and distribution between estimated HR and ground-truth HR across all the subjects in the MMSE dataset. Figure 10 illustrates the errors for 2D-CAN and MTTS-CAN on the AFRL Dataset by participant.

Figure 3: Architecture details of TS-CAN and MTTS-CAN

Figure 4: Architecture details of Hybrid-CAN and MT-Hybrid-CAN

Figure 5: Architecture details of 3D-CAN and MT-3DCAN

# 2D-CAN

# 3D-CAN

# Hybrid-CAN

Figure 6: AFRL Dataset: Estimated HR and Gold-standard HR reference measurements in (beats-per-minute) and the corresponding Bland-Altman plots from 2D-CAN, 3D-CAN and Hybrid-CAN. All results computed for 30-second time windows.

Figure 7: AFRL Dataset: Estimated HR and Gold-standard HR reference measurements in (beats-per-minute) and the corresponding Bland-Altman plots from MT-Hybrid-CAN and MTTS-CAN. All results computed for 30-second time windows.

## 2D-CAN

## 3D-CAN

## Hybrid-CAN

Figure 8: MMSE-HR Dataset: Estimated HR and Gold-standard HR reference measurements in (beats-per-minute) and the corresponding Bland-Altman plots from 2D-CAN, 3D-CAN and Hybrid-CAN. Note the following files were removed due to unreliable reference measurements: F006-T11, F013-T08, F013-T11, F014-T01, F014-T08, F014-T10, F014-T11, F015-T11, F022-T10, M013-T11

Figure 9: MMSE-HR Dataset: Estimated HR and Gold-standard HR reference measurements in (beats-per-minute) and the corresponding Bland-Altman plots from MT-Hybrid-CAN and MTTS-CAN.

Figure 10: Box Plot of Errors for 2D-CAN and MTTS-CAN on the AFRL Dataset by Participant.