[Reviews · NeurIPS 2020]

Review 1

Summary and Contributions: The authors present a multi-task temporal shift convolutional attention network (MTTS-CAN) to address the challenges of privacy, portability, and precision in contactless cardiopulmonary measurement. MTTS-CAN utilizes the benefits of temporal shift module, attention mechanism and multi-task learning. Temporal shift module is used to better capture the temporal information without additional overload. The attention mechanism is used for better signal source separation. Pulse and respiration are jointly trained to learn better intermediate representation. The method has been evaluated on two public datasets. The authors have also developed code base for efficiently executing tensor operations. They report 150 fps.

Strengths: The problem itself is an interesting one to approach, obtaining physiological measurement through camera data. Methodology: 1. Averaging the frames in the appearance branch in the proposed network is intuitive as it removes the computation load of 3D convolutions. 2. 3D convolutions in the motion branch are replaced by a temporal shift module, which has been a promising alternative for 3D convolution operations. 3. Attention mask provided by appearance branch to motion branch is an interesting approach to alleviate noise. 4. As both pulse and respiration are linked, learning them together looks like an ideal choice. Experiments: 1. Comparison with existing methods 2. Extensive ablative study on performance and latency for the proposed models. 3. Code shared.

Weaknesses: 1. From the prior works, it is clear that the two branch structure, temporal shift module, attention mechanism are already available. The authors have shrewdly used these modules for an interesting problem. Proposing new methods for temporal shift module would have still added value to the paper, as there is a large performance gap between 3D and temporal shift methods. However, it is understandable that this is the common performance/latency tradeoff. How about model compression techniques like (knowledge distillation, model pruning ?). 2. Effect of attention module on the temporal operation is not clearly explained. What is the effect of the module on handling the noise? 3. In motion cases, how will the appreance branch help the motion branch ? How will the images look ? Would it be better to use temporal data here rather than direct mean ? 4. FPS comparison in table 1 would be really helpful. 5. Why in some cases TS-CAN is better than MTTS-CAN?

Correctness: Yes. The authors have good knowledge about the area of research and the proposed methods and claims are correct.

Clarity: Paper is well written and easy to follow.

Relation to Prior Work: The authors have clearly discussed how their work is different from related works.

Reproducibility: Yes

Additional Feedback:


Review 2

Summary and Contributions: The paper is a development to improve the (1) accuracy, (2) latency, and (3) computational burden of vital sign estimation via contactless/camera-based video monitoring of the face. Estimation is performed via a temporal shift convolutional attention network (as the title implies). This is a sensible approach to a well-established and important field of patient vital sign estimation & monitoring. The algorithmic components have a well-established connection with the salient physiological phenomenon under consideration. The new proposed model attempts to improve over previous baselines by incorporating dynamics in the physiology that are currently under-utilized in modeling and estimation. A multitask approach (to the two vital signs) allowed a halving of computational burden with little affect (positively or negatively) on predictive accuracy compared to a single-task model. Improvements in this domain are of value to clinical domains such as the NICU and e-health. This paper (as is) would be of significant interested to ML practitioners in the field as a solid step in the right direction.

Strengths: This paper is better than any of the ~1/2 dozen vital-sign-inference journal papers I’ve reviewed over the last two years. The physiological grounding is strong and they selected an algorithm that better incorporates key / high-priority physiological mechanisms. They explain the trade-offs they were attempting to balance, which places the paper far beyond just applying a new algorithm for the sake of a new algorithm. They explained the clinical domain well (Sections 1 and 2 are very good, with appropriate focus and structure, which should not be taken for granted). Novelty: This field is well-established and the general methods are well established. This paper does not revolutionize either of these (nor does it claim to) but it is a very sensible approach to advance the state of the art of the field. It certainly stands out in terms of well-explained technical choices with regards to its novel contributions. As a whole, the paper has sufficient novelty of methods and results to be of interest. Empirical evaluation: The empirical evaluation is very standard for the field (as good as any I have seen). It is also better than most since it tests models for accuracy across data sets, see lines 277-281, (which should be done more often, and demonstrates the authors’ appreciation of the inductive challenges to clinical science). Empirical evaluation could be improved by breaking down the performance metrics on a by-patient The video is excellent and compelling. (But please clarify the contract respiratory sensor in the figure label.)

Weaknesses: This article was good, and our field would benefit from having this article elevated. A quibble: Was this article submitted to a COVID-19-specific theme session? If not…I’m not really sure that all the references to COVID are important or add much to the relevance of the contribution. Non-contact was important before COVID, it will be important after (if there is an after). But the COVID *focus* seems strange given all the other important needs and applications for non-contact vitals. The paper stands well on its own without COVID. Just a thought. One actual weakness (and this is a general issue in healthcare papers so this paper isn’t alone or rare in this regard): The paper takes results from dozens of patients and dozens of videos…and summarizes performance into a single performance metric. This masks several aspects that are essential to the evaluation of clinical performance including: [1] The accuracy of estimation on individual patients. [2] The accuracy of estimation in specific clinically-relevant vital sign ranges. A key issue with these algorithms is being able to claim that they would work on the patient groups of interest during the physiological periods of interest (which are typically more extreme and harder to estimate). The paper would be significantly improved by adding the following graphs to the appendix: [1] A plot of patient index (x-axis) vs the patient’s ground-truth vital sign value distribution (y-axis). The raw values would be too busy to be readable, so summarizing the distribution via the quantiles or a violin plot would be fine. A violin plot might be too bulky (horizontally). Plotting the quantiles (ex. .1, .25, .5, .75, .9) would be space efficient an communicate the patient-specific ranges of each ground truth vital sign. (Sorting the patient index by median vital sign value would likely improve readability). [2] The same plot above, but replacing the Y-values with the estimation errors for each patient. This would clarify whether the algorithm performed with similar accuracy across all patients. (Sorting the patient index by median error value would likely improve readability). [3] A Bland-Altman plot on estimated vs gold standard values (or something performing a similar function). Bland Altman plots typically include the error distribution…but it might be visually easier to just “bin” the error values into by the value of the gold standard (ex. HR 30 to 200 by 5’s, Resp rate 0 to 45 by groups of 2 or 3). This would clarify whether the algorithm performed with similar accuracy across different ranges of the underlying value. These plots are essential to evaluate predictive biases that would be missed by the aggregate statistics. (I actually drew a picture to help show the desired plots but there was nowhere to upload the picture.) The paper is good and will be better than many papers without this addition. But the paper would be much better with these additions.

Correctness: The claims and methods are correct and based on well-established standards of the field. The empirical methodology is correct *in describing a significant improvement to average estimation performance*. My only objection to the correctness of the paper is the one listed above, which would constitute a minor edit. Otherwise the paper does a very good job verify it’s claims and clarifying key experimental decisions.

Clarity: Yes. The structure is clear and it is written with a strong appreciation of (1) the important physiological mechanisms, (2) the implications of physiology on clinical actions, and (3) the technical and clinical limitations of the ML application. This paper is a good example for what applied healthcare ML publications should emphasize (ex. connections of algorithm to underlying physiology). The healthcare ML field would benefit from having this paper elevated as a positive example. On length: The references begin at page 9 but would be ~7 or 8 pages without header, figures, tables, equations. So it meets the length criterion. Minor edits / quibbles: The following are writing issues…and do not hamper the scientific contributions. Lines 3-4: There are more reasons (esp. beyond COVID challenges) why contactless vitals is important. Line 6: “Objective measurement…is challenging without contact.” - It’s really challenging even with contact! Line 7: “On-device”…please specify the device. Line 10: ARM CPU – please spell out on first use. …Also things like ER, NICU (line 40), etc. Line 13: “reductions in errors”  qualify the errors. Bad vitals can result in several types of errors. You mean an inaccurate estimate of the vitals themselves. General issue: latency…..It seems that latency is only an issue in that it hampers precision in estimating waveform periodicity. But early on its presented as if latency is its own clinical issue, instead of just a means towards predictive accuracy. Does a latency of 6ms vs 48 ms matter for any clinical aspect other than predicting the vitals themselves? Is there some other function mapping latency to other clinical challenges of interest? Latency goal is clarified 291-295 but should have been done earlier. 33-34: On symptoms: Do these symptoms manifest in the vital signs in any but the most extreme cases? If the cardiopulmonary symptoms are severe, do you need vital sign measurements to diagnose? 37: The term “cardiovascular protection” is unclear. 54: “This standard has not been obtained.” In non-contact? Should clarify that you mean “in a non-contact approach.” …More importantly, this isn’t really true…non-contact approaches could claim that a contact approach is substantially different when applying for approval and therefor the comparison is unfair. Should have a one-sentence explicit explanation of the physiology that the camera is picking up (near the beginning or at latest around line 77) about the mechanism. Something about the pulsatile flushing of blood through the face that is detectable to cameras but not to the human eye. 82: “illumination changes” – (e.g., ambient lighting, over-saturation, etc.) Figure 1 – really like this! Line 90: “takes” is an ambiguous word. “Requires”? “is limited to”? Clarify the nature of the algorithmic bottle neck. Line 110-115: Again, seems like a random connection to COVID. Is this a COVID session? Even without COVID this work is relevant and stands on its own. Line 135-138: Good example of highlighting the physiological interactions captured. Very nice and helpful. Important for more papers to have descriptions like this. List starting at Line 147: …it seems that only “3” is an attribute required for state of the art clinical performance. Whereas “1” and “2” are means to achieve “3”. …a quibble but it seems like a conflation of “sensible tactics to achieve requirements” versus the actual requirements themselves. Line 158/159, 163/164, 184-188 – really like these descriptions and writing. 212:-2:14 – Please clarify a little more. Why not use ECG as gold standard throughout? Line 265 “(same folds as 9)” – nice tidbit/detail. I wish more papers connected their approach to previous work like this. Line 272: “We observe …both tasks.” - A nice technical insight. Appreciated! Lines 277 – 281: Very good. I wish more papers would try approaches like this. 318-327: This is an interesting nuance. Better than the “multitask-is-always-better-so-I-won’t-check” stuff that’s in a lot of papers. 337-338: This statement is a little over-done and only believable in regard to avoiding contac tot avoid infection. Otherwise, the large number of long-standing non-contact clinical needs is an strong as ever…and not particularly highlighted by COVID-19. A suffering neonate or burn victim highlights this need well before COVID. 345-347: The resource constrained issue is very important and should be brought to the front of the article too….preferably in the abstract if there’s space. Watch of capitalization in Latex References. Ex. “ct” or “covid” in ref 25. Use the Latex “{}” to fix.

Relation to Prior Work: Yes. This is done clearly and accurately with proper attention to clinical needs and key preceding work. They're certainly referencing the work that I'm familiar with in the field.

Reproducibility: Yes

Additional Feedback: This is a very good paper and should be elevated by our field as a great example of progressing a well-developed field. The paper should be accepted. The requested plots are minor edits that should be added to the appendix.


Review 3

Summary and Contributions: The paper follows a well written path of clear understanding and idea of the concept. The authors have made sure they cover all the loopholes which were present in similar past papers keeping in mind the importance of real time low-latency system, which is a must when talking about any technology dealing with tele-health or remote health assessment. The authors provide a simple looking yet complex technique to obtain on-device contactless vital measurements related to cardio-pulmonary measurements. They use computer vision to capture subtle differences in the subject’s body and figure out the needed wave forms. The introduction to a multi-task temporal shift convolutional attention network not only uses the advantages of a 3D CAN over 2D CAN, but also features the implementation of hybrid CANs making it a far better choice while dealing with multiple waveforms (blood volume pulse and respiratory wave) simultaneously in real-time. Authors validated the new-found model on large benchmark datasets to find their model leads to a significant (20% to 50%) reduction in errors when generalized across all datasets. This is a very substantial claim which they have provided proofs for in their experimental and result section. The authors through this paper have provided an efficient and precise way of performing on-device real-time spatial-temporal modeling of vital signs, in addition to proving it to be a state-of-the-art performance tested across standard dataset and provided a means to implementation of code tensor operation with minimum latency.

Strengths: The strengths of the paper are provided below along with some explanations: • Implementing Shafer’s DRM: The authors have used Dichromatic Reflection principle by Shafer, which states that under all illumination and viewing geometries the spectral reflection function is described by weighted sum of two functions of constant interface reflectance and body reflectance. This helps the attention network to improve its precision. Using this principle also helps in removing visual noise by separating the body illuminance. • Temporal Shifting: Authors have used temporal shift module which splits an input tensor into three chunks across the input channel dimension. They then shift the first chunk to left (in past) and one chunk to right (in future) which captures the temporal information for that point. Though this also introduces noise but as per their calculations in further stages, they have • TS-CAN overcoming difficulties dealing with 3D CAN: The authors checked on processing the algorithm with 3D CAN, and though it provided with better accuracies compared to 2D CAN, the time taken was not acceptable. Their use of hybrid TS-CAN which could use the advantages of both 3D-2D CAN had better tradeoff of time and accuracy compared to all other options. TS-CAN not only removed the complex convolution operations from the architecture, but also gave authors some flexibility to add different components. • Accuracies for multi-task models remains constant: In the results section, authors evaluated the multi-tasking version of TS-CAN as well as Hybrid-CAN observing no accuracy difference. This was in a way remarkable as the increasing computation complexity of hybrid version and multi-tasking versions did not affect the accuracy in any way.

Weaknesses: The weakness of the paper is provided below along with some explanations: • Heavy reliance on attention network: The authors focus on understanding or realizing the illumination from subject’s body part and skin which requires the attention network to have access to these information. Also, the implementation of Shafer’s DRM helps in differentiating the foreground and body illumination, however the need to label them still exist. Therefore, the approach provided here has a high reliance on use of attention networks to present the subject and details about the subject. • Need for a higher resolution optical equipment: As the algorithm focusses on optical illumination and reflection, there is a need to focus and capture pixels with high density of clarity and information. Therefore, even though the accuracy of the algorithm seems good, it to a large extent depends high resolution camera picking up the optical information. • No explanation on external effects: Human beings tend to cover body with different types of clothes that may come in various sizes and layers. The explanation of authors depends on obtaining ques from subject’s body based on changes in body position, raising of head or chest, and even changes in color variation of skin which would be difficult with layers of clothes on subject and authors do not provide any option towards a solution to this. • Firefly-RK3399: Firefly RK 3300 is a 6-core 64 bit high performance platform. This is a high performing collection and the algorithm with such complex computations might work faster in such combination. Authors should try implementing the algorithm and run over different lesser strong processors to see how the algorithm works. • Minor comments: o The topic is about on-device, but the presentation of this point is not enough. The author just uses a small section to explain the on-device evaluation. o The position of Fig.1 and Fig.2 should be close to the corresponding explanatory text, which helps readers understand them better. Especially for Fig.1, there is a comparison for four models, but it is hard to understand the content of it just with the simple legend. Or, the author can add the legend with more details (what is A, B, C, and D?!) to make it easier to understand.

Correctness: Authors have claimed using computer vision to get contact less vital measurements from subjects. The Authors claim to get reduced errors and increased accuracy using their provided MTTS-CAN algorithm and they have also provided the proofs of their claim by comparing the algorithm over standard datasets with other benchmark algorithms. The use of certain techniques while explaining the algorithm makes sense. Like using Shafer’s DRM principle in combination to CAN could help leverage the fact that the authors are focusing on layering the subject out of the background and running the main process on that.

Clarity: The paper is well written technically, however lakes some fundamental explanation of reasons that the authors chose to select a few options. The authors have not explained well on how does the illumination and surface reflection help in understanding the changes in the skin tone or a matter of fact understand how the changes on color or skin helps understand the respiratory or blood waveform. However, authors have explained how they are using the fact that there is a correlation between two waveforms and how they have leveraged that to obtain their MTTS-CAN algorithm. They have also explained well how their interpretation of Shafer’s DRM into understanding the different parts of background and body as well as understanding the illumination of surface from body helps us recognize the subtle differences in skin that could be leveraged to understand their correlation with measurement of vitals.

Relation to Prior Work: One major contribution that authors provide here is the fact that they figured out the correlation between respiration rate and blood volume pulse and used it to predict the independent waveforms together simultaneously. The prior work mentioned in the paper discuss the prediction of such waveforms independently with different methods, however this approach lets us estimate the waveforms together. This helps to minimize the over-head in addition to maximizing the output capacity of the algorithm.

Reproducibility: Yes

Additional Feedback:


Review 4

Summary and Contributions: This paper looks at the use of temporal shifting, attention, and multi-task inference with convolutional networks as a means of facilitating high FPS inference of cardiopulmonary waveforms from video. They compare these approaches to 3D convolutions and a variety of approaches that mix and match the above. While the individual techniques are not novel, their combination for the purposes of gathering vital signs from video is particularly timely granted the current global pandemic and need for remote monitoring. The authors demonstrate solid results on standard benchmarks with their approach.

Strengths: The authors employ several techniques that have been demonstrated elsewhere to improve performance for video based tasks or computer vision tasks in general. They benchmark against standard CNN based approaches as well as non-deep learning based methods, and measure key metrics of quality (MAE, SNR, etc…) and inference time (ms). The authors are particularly thoughtful about the deployment aspect of their system, and make a point of getting it compiled and running on a Firefly-RK3399.

Weaknesses: I think this papers biggest weakness is that I’m not convinced it is significantly better than a 3D-CNN? The 3D-CNN can only run at ~20 FPS compared to the MTTS-CAN at ~160 FPS, but isn’t 20 FPS more than enough for measuring blood pressure and respiratory rate. I don't disagree with the authors that a higher sampling rate is important for calculating variability measures at a reasonable resolution. However, I'm questioning the fundamental problem requiring such measures. We simply don't use that information clinically.

Correctness: The methodology is correct, and well supported by provided code and documentation.

Clarity: Yes

Relation to Prior Work: Yes

Reproducibility: Yes

Additional Feedback: I'm not convinced that the problem is relevant. Their approach has a faster FPS than a 3D-CNN, but for the problems they're tackling this seems sufficient? I think re-writing to focus on the implementation on flexible, easily deployable hardware might improve the manuscript (or looking at power requirements, etc...).

[Author Response · NeurIPS 2020]

We would like to thank the reviewers for their extensive and constructive feedback and are glad they found our work a *strong contribution*. We have made an effort to answer all their comments and will update our paper accordingly.

**R2 & R5: Why is a high frame rate important?** The standard sampling rate for contact pulse and respiration sensors is between 100-256Hz. This is because derivative information is often computed from these waveforms. For example, for heart and respiration rate variability measurement 20 Hz is not sufficient because peak timings are too imprecise.
**R2 & R3: How does the illumination and surface reflection help in understanding the resp or pulse waveforms?** The volumetric changes of blood in the surface of the skin cause changes in light absorption (↑ volume of hemoglobin = ↑ light absorption). This in turns affects the amount of visible light reflected from the skin, this is the source of the photoplethysmogram. The mechanical force of blood pumping around the body also causes subtle motions and these are the source of the ballistocardiogram. These color and motion changes in the video help us extract the pulse signal and heart rate frequency. Respiration is captured mostly via motions of the torso as the subjects breathes in and out. But, the principle of respiratory sinus arrhythmia also means that the color changes of the skin pixels due to blood flow also contain some respiration information. Separating these motions and color changes of interest from specular reflections and other motion noise is the primary task of camera-based physiological measurement.

**R1**: 1) We believe the computational benefits offered by our approach are a helpful contribution to the research literature. Applying knowledge distillation and model pruning to Hybrid/3D-CAN are definitely interesting lines of investigation but are beyond the scope here. 2) The attention module learns higher weights for skin regions in the pulse task, and torso regions in the resp. task. This boosts the SNR in the motion branch because changes from other sources can be more effectively ignored. Fig. 3-B shows that the temporal shifting in the motion branch can introduce more signal but also more noise, therefore the attention module is even more important. 3) Hybrid-CAN uses a 2D attention branch with an averaged frame whereas 3D-CAN uses a 3D attention branch with all the frames. We have shown in Table 1 that Hybrid-CAN was able to achieve a MAE of 1.12 while 3D-CAN had a MAE of 1.18. We experimented using the middle frame and last frame as the input to the 2D attention branch. We found that the mean frame provided the best results. Since the 2D branch only requires a single frame as input (the average frame), the generated attention mask can be shared with all the frames (10 in our case) in the motion branch, helping significantly reduce computation with little impact on accuracy. 4) TS-CAN sometimes performs better than MTTS-CAN because we train two separate "dedicated" TS-CAN models, one for pulse and one for resp. However, we only train one multi-task (MTTS-CAN) model which has to share the weights for two tasks and is much more computationally efficient but is not always as accurate.

**R2**: 1) We agree with that our approach could have applications beyond telehealth monitoring during a pandemic. We wrote our introduction to resonate with the current situation, but we agree it would be good to reduce this emphasis and highlight more applications in other domains. 2) We will add a breakdown of the performance by participant in the supp. material. The st.dev. across subjects was (MTTS-CAN): HR MAE: 1.15 BPM (AFRL) and 4.82 BPM (MMSE) and respiration MAE: 1.62 breaths/min (AFRL), these combined with the violin and Bland-Altman plots will be helpful. We agree about evaluation on patient groups of interest (e.g., COVID and A.Fib.) but a full clinical validation was beyond the scope of this work. Generally speaking we expect that improvements in accuracy on healthy people will also help with performance on patients. 3) Vital sign measurement is still imperative even if the cardiopulmonary symptoms are severe. Scalable and accessible health monitoring has many applications and could help democratize care.

**R3**: 1) The spatial resolution of the videos does not need to be very high (658x492 for AFRL) and we actually downsample the face regions to 36x36 pixels, which is quite effective (Chen & McDuff, 2018). Studies have shown that off-the-shelf webcams are sufficient for physiological measurement. We would argue that high resolution cameras are not necessary. 2) The reason we proposed the use of multi-task learning and tensor shifting is to enable our network to run on edge devices in real-time while maintaining a sufficient frame rate and state-of-the-art accuracy. Our goal in Sec. 3.2 was to provide an explanation of how to realize on-device real-time temporal and spatial modeling. Table 1 and Figure 3-A capture these results. 3) The Firefly RK-3399 has a CPU with 2 Cortex A72 (1.8Ghz) and 4 Cortex-A53 cores. This is equivalent to the CPU in the Snapdragon (Snap) 650 Mobile Platform[1]. Snap 650 is a mid-to-low-end mobile platform that Qualcomm launched in 2015. Snap 650 achieves a score of 275 in single-core evaluation and a score of 830 in multi-score evaluation[2]. The modern Snap 865 achieves a score of 903 in single-core evaluation and a score of 3304 in multi-score evaluation. To contextualize this, Snap 650 has a little worse performance than the Apple A9 chip used in the iPhone 6s. Thus, we believe that the computing platform we tested on is a fair benchmark for the class of mobile devices we are targeting. 4) Clothing and facial hair do obscure the skin and can affect the measurement of the pulse signal, indeed a limitation of optical measurement via PPG. Skin type can also have an effect, we coded the skin type and gender of subjects in AFRL using the Fitzpatrick scale. The results (MTTS-CAN) are as follows (Skin Type-MAE): I-2.37 BPM, II-2.62 BPM, 1.77-BPM, IV-0.06 BPM. (Gender-MAE): Men-1.34 BPM, Women-3.92 BPM.

**R5**: We will emphasis why our method makes the deployment of camera-based vital measurement more tractable given that the lower computation budget requires less power consumption per inference, we appreciate that suggestion.

## Footnotes

[1] https://www.qualcomm.com/products/snapdragon-650-mobile-platform  [2] https://browser.geekbench.com/


[Meta-Review · NeurIPS 2020]

All four knowleageable reviewers are positive about the paper. The rebuttal further confirmed their thoughts. I particulary agree with R2 that it is not groundbreaking but it is a very sensible approach to advance the state of the art of the field. There are some weak points raised by the reviewers; the authors are encouraged to further improve the paper quality by addressing them.